# Provision of Psychotherapy during the COVID-19 Pandemic among Czech, German and Slovak Psychotherapists

**DOI:** 10.3390/ijerph17134811

**Published:** 2020-07-04

**Authors:** Elke Humer, Christoph Pieh, Martin Kuska, Antonia Barke, Bettina K. Doering, Katharina Gossmann, Radek Trnka, Zdenek Meier, Natalia Kascakova, Peter Tavel, Thomas Probst

**Affiliations:** 1Department for Psychotherapy and Biopsychosocial Health, Danube University Krems, 3500 Krems, Austria; elke.humer@donau-uni.ac.at (E.H.); christoph.pieh@donau-uni.ac.at (C.P.); martin.kuska@donau-uni.ac.at (M.K.); 2Clinical and Biological Psychology, Catholic University of Eichstätt-Ingolstadt, 85072 Eichstätt, Germany; antonia.barke@ku.de (A.B.); bettina.doering@ku.de (B.K.D.); katharina.gossmann@ku.de (K.G.); 3Science and Research Department, Prague College of Psychosocial Studies, 14900 Prague, Czech Republic; trnkar@volny.cz; 4Olomouc University Social Health Institute (OUSHI), Palacky University Olomouc, 77111 Olomouc, Czech Republic; zdenek.meier@upol.cz (Z.M.); natalia.kascakova@oushi.upol.cz (N.K.); peter.tavel@upol.cz (P.T.); 5Psychiatric-Psychotherapeutic Outpatient Clinic, Pro mente sana, 81108 Bratislava, Slovakia

**Keywords:** psychotherapy, COVID-19, public health, fear of infection, remote psychotherapy

## Abstract

Psychotherapists around the world are facing an unprecedented situation with the outbreak of the novel coronavirus disease (COVID-19). To combat the rapid spread of the virus, direct contact with others has to be avoided when possible. Therefore, remote psychotherapy provides a valuable option to continue mental health care during the COVID-19 pandemic. The present study investigated the fear of psychotherapists to become infected with COVID-19 during psychotherapy in personal contact and assessed how the provision of psychotherapy changed due to the COVID-19 situation and whether there were differences with regard to country and gender. Psychotherapists from three European countries: Czech Republic (CZ, *n* = 112), Germany (DE, *n* = 130) and Slovakia (SK, *n* = 96), with on average 77.8% female participants, completed an online survey. Participants rated the fear of COVID-19 infection during face-to-face psychotherapy and reported the number of patients treated on average per week (in personal contact, via telephone, via internet) during the COVID-19 situation as well as (retrospectively) in the months before. Fear of COVID-19 infection was highest in SK and lowest in DE (*p* < 0.001) and was higher in female compared to male psychotherapists (*p* = 0.021). In all countries, the number of patients treated on average per week in personal contact decreased (*p* < 0.001) and remote psychotherapies increased (*p* < 0.001), with more patients being treated via internet than via telephone during the COVID-19 situation (*p* < 0.001). Furthermore, female psychotherapists treated less patients in personal contact (*p* = 0.036), while they treated more patients via telephone than their male colleagues (*p* = 0.015). Overall, the total number of patients treated did not differ during COVID-19 from the months before (*p* = 0.133) and psychotherapy in personal contact remained the most common treatment modality. Results imply that the supply of mental health care could be maintained during COVID-19 and that changes in the provision of psychotherapy vary among countries and gender.

## 1. Introduction

The outbreak of the novel coronavirus disease-2019 (COVID-19) causes changes in the provision of mental health care in many countries [1]. In an attempt to reduce the risk of infections, many psychotherapists reduce or even quit the provision of face-to-face psychotherapy and simultaneously try to replace psychotherapy in personal contact with remote psychotherapies to provide mental health care at a safe distance [2,3,4]. At the same time, mental health problems increase not only among infected patients and relatives, but also in the general population [1,4,5]. Thus, there is a high need for timely mental health care during and after the COVID-19 pandemic [2]. However, the common treatment format of psychotherapy, i.e., face-to-face personal contacts, poses the risk of transmitting the infection between psychotherapists and patients and face-to-face personal contacts in general should be reduced. The obvious solution to providing mental health care during a pandemic is to change the route of delivery and to implement psychotherapy remotely via telephone or internet [1,6,7].

Prior to the COVID-19 outbreak, the limited acceptance of remote psychotherapy by healthcare providers featured among the most important barriers for the implementation of psychotherapy provided from distance [8]. Concerns regarding remote psychotherapy, such as the impossibility to develop a therapeutic alliance or reach an accurate diagnosis from a distance [8,9], were often raised by psychotherapists. Patients on the other hand, seemed to show more positive attitudes toward remote psychotherapy [10,11]. In patients, no difference in patient-rated alliance scores between psychotherapy in personal contact and psychotherapy from distance was observed [12]. Also, research regarding the effectiveness of remote psychotherapy showed promising results [13,14], revealing comparable outcomes of providing psychotherapy via telephone or internet as compared to psychotherapy in personal contact [15,16,17].

The delivery of remote psychotherapy may also be impeded by regulatory barriers to remote psychotherapy, which were loosened in many countries in response to the COVID-19 outbreak [18]. However, differences among countries exist in lockdown measures and official guidelines with regard to remote psychotherapy. The following outline provides the description of measures related to the COVID-19 outbreak in the three countries under investigation in this study.

In the Czech Republic (CZ), a prohibition on visits to social facilities and hospitals was imposed from 9 March 2020, i.e., 9 days after the first three patients were positively diagnosed with COVID-19. After that, the total closure of the national borders (13 March 2020), the general closure of services and retail sale, with the exceptions of grocery shopping (14 March 2020), and nationwide curfew, with the exception of ways from home to work and back (16 March 2020) followed. From 19 March 2020, the obligation of everyone to wear surgical or homemade masks covering the mouth and nose was imposed. The nationwide curfew ended on 24 April 2020. Before the COVID-19 outbreak, no remote psychotherapy was covered by the health insurance companies. During March 2020, remote psychotherapies started to be fully covered by the health insurance companies to psychotherapists that have a contract with some of them. The coverage of remote psychotherapies ended by various dates during the second half of May 2020, depending on the decision of each health insurance company.

Germany (DE) went into lockdown beginning 22 March 2020, banning public gatherings of more than two people, closing schools and non-essential businesses, and urging residents to stay at least 1.5 meters away from each other [19]. The nationwide curfew ended on 6 May 2020; however, in Germany, the measures were largely a matter of the federal states and many differences within Germany existed. Concerning coverage of remote psychotherapy by health insurances before COVID-19, psychotherapists in Germany could conduct up to 20% of their overall treatments per quarter remotely with an upper limit of 20% of patients per quarter receiving remote psychotherapy as only treatment format. Some psychotherapeutic services, however, such as the initial interview or diagnostics were only covered when conducted face-to-face. Soon after the start of the lockdown up to June 1 2020, the limit concerning the amount of remote therapy was suspended and additional services were reimbursed when conducted remotely.

Slovakia (SK) started its state of emergency on 16 March 2020 with closing non-essential stores, banning public gatherings of more than two people, closing schools and non-essential businesses, and urging residents to maintain social distance of 2 meters and to wear mandatory facemasks. After recommendations of the Ministry of Health, only the necessary medical care was provided, including psychiatric and psychotherapeutic care, which was recommended to be provided mostly online, but without clear instructions to what extent it will be reimbursed by health insurers. Free movement was limited within one district during the Easter holiday from 8 to 14 April 2020. Exceptions from these restrictions were business activities and necessary medical visits. The ban of movement ended on 14 April 2020 and the lifting of the quarantine followed in four stages. Kindergartens and schools were opened in only a limited way on 1 June 2020. The state of emergency ended on 16 June 2020.

At the time of the survey (6 May 2020 until 20 May 2020 in CZ, 19 May 2020 until 28 May 2020 in DE, and 8 May 2020 until 22 May 2020 in SK), no curfews existed any longer in the participating countries.

A further factor that might affect the format in which psychotherapy is provided during the COVID-19 pandemic might be the fear of COVID-19 infection during psychotherapy in personal contact. As different countries are affected differentially by the COVID-19 pandemic, and media reports as well as legal restrictions differ among countries, this fear might differ among countries. Furthermore, women have reported higher levels of health-related fears than men [20] and a recent study showed that fear was reported more frequently by women (67%) than men (33%) when asked which emotions they felt at the time of the COVID-19 pandemic [21,22]. Gender differences were also revealed to exist with regard to the prevalence of posttraumatic stress symptoms after the COVID-19 outbreak in China, with stronger emotional changes in women [23].

The aim of the present study was to evaluate the following research questions (RQs) with regard to psychotherapists’ self-reports in CZ, DE, and SK.

RQ 1: Are there differences in fear of COVID-19 infection during psychotherapy in personal contact between CZ, DE, SK and male vs. female psychotherapists?

RQ 2: Have the numbers of patients treated on average per week during COVID-19 changed when compared to the months before COVID-19 for the total sample as well as for CZ, DE, and SK separately?

RQ 3: Are these changes influenced by treatment format (personal contact, telephone, and internet), gender, and country?

RQ 4: Are these changes influenced by the fear of COVID-19 infection during psychotherapy in personal contact?

## 2. Materials and Methods

### 2.1. Study Design

An online survey designed in Research Electronic Data Capture (REDCap) [24] was sent via e-mail to licensed psychotherapists in CZ, DE, and SK. All participants gave informed consent by agreeing to the data protection declaration before the start of the survey.

In CZ, the psychotherapists were contacted through the email list of the Czech Association for Psychotherapy (https://czap.cz/), a Czech national association joining a high number of Czech psychotherapists.

In DE, all e-mail addresses were gathered from the publicly available directories of four different regional and national psychotherapeutic associations. In these directories, the associations publish the contact information of all licensed psychotherapists who gave their consent to such publication.

In SK, emails with information about an online survey were sent to the chairman of the Slovak Psychotherapeutic Society and then to the chairmen of special psychotherapeutic societies and then sent from these sources to psychotherapists via e-mail lists.

The psychotherapists who were interested to participate filled an online questionnaire. The data were then automatically sent to a central data set.

In CZ, the survey was open from 6 May 2020 until 20 May 2020, in DE from 19 May 2020 until 28 May 2020, and in SK from 8 May 2020 until 22 May 2020. For CZ, this was about 7 weeks after lockdown measures were initiated and about 2 weeks after restrictions began to be lifted. For DE, this was about 8 weeks after lockdown measures were initiated and about 2 weeks after restrictions began to be lifted. For SK, this occurred 7 weeks after lockdown measures were initiated and about 3 weeks after restrictions began to be lifted. The survey was conducted during the week when the second phase (out of four) of lifting restrictions began (e.g., permission for church services, weddings, hairdressers, short-term accommodations without boarding). Kindergartens and schools were still closed and were opened only in a limited way as of the 1 June 2020.

### 2.2. Measures

Psychotherapists were asked about the number of patients treated on average per week in personal contact, via telephone and via internet in the months before the COVID-19 situation (retrospectively) as well as since the COVID-19 situation. These numbers were set to 0 for psychotherapists not treating before or during the COVID-19 situation.

Additionally, they were asked to rate their fear of becoming infected with COVID-19 during psychotherapy in which they are in personal contact with patients on a slider ranging from 0 (“not at all”) to 100 (“extreme”).

### 2.3. Statistical Analyses

Statistical analyses were conducted with SPSS version 25 (Inc, Chicago, IL, USA).

To evaluate differences in sociodemographic characteristics, univariate analysis of variance (ANOVAs) and chi-square-tests were conducted.

Statistics for RQ 1: To investigate fear of COVID-19 infection, a univariate 3 x 2 ANOVA with the factors country (CZ, DE, SK) and gender (female, male) was performed.

Statistics for RQ 2: A t-test for dependent samples was performed to investigate whether the total number of patients treated on average per week during COVID-19 differed from the number of patients treated on average per week in the months before the COVID-19 situation. Further t-tests were conducted to investigate a possible change in the number of patients treated during COVID-19 compared to the months before for each format (personal contact, telephone, and internet), separately. Also, paired t-tests were performed to investigate whether the decrease in the patients treated in personal contact during COVID-19 as compared to the months before, differed from the increase in the number of patients treated remotely (telephone + internet). All tests were performed for the total sample, as well as for each country (CZ, DE, and SK) separately.

Statistics for RQ 3: Mixed ANOVAs (RM-ANOVAs) were performed to investigate whether changes in the number of patients treated on average per week (during COVID-19 vs. months before) interacted with the three treatment formats (personal contact, telephone, and internet), the three participating countries (CZ, DE, SK) and gender (female, male). In this RM-ANOVA, the number of patients treated on average per week was the dependent variable. There were two within-subject factors, the first was “change” (two levels: during COVID-19, months before COVID-19) and the second was “format” (three levels: personal contact, telephone, internet). There were two between-subject factors, i.e., “country” (three levels: CZ, DE, SK) and “gender” (two levels: female, male). All main effects (ME) and interaction effects (IE) were examined. The Greenhouse-Geisser corrected values are presented.

Statistics for RQ 4: Pearson correlation analyses were performed to reveal an association between the fear of COVID-19 infection during psychotherapy in personal contact and the number of patients treated in personal contact, via telephone or via internet during the COVID-19 situation. Also, changes in the provision of psychotherapy in personal contact, via telephone and via internet during COVID-19, as compared to the months before COVID-19, were correlated with the fear of COVID-19 infection. These changes were calculated as the number of patients treated on average per week during COVID-19 minus the number of patients treated on average per week in the months before COVID-19 for each treatment format.

All tests were performed two-tailed with a significance value of *p* < 0.05. Bonferroni corrections were applied for the pairwise-post-hoc tests.

## 3. Results

### 3.1. Participant Characteristis

In total, 338 psychotherapists participated (CZ: 112, DE: 130, SK: 96). Their mean age was 46.70 (SD = 10.68) years, which differed among participating countries (F(2; 2450.2) = 24.493; *p* < 0.001; Table 1). German psychotherapists were the oldest, differing from CZ and SK with *p* < 0.001. No age difference between CZ and SK (*p* = 0.806) was observed. On average, 77.8% of the psychotherapists were female and the gender distribution did not differ among countries (x^2^(2) = 3.067; *p* = 0.216). Similar to differences in age, the average years in profession (M = 10.75, SD = 9.98) differed among countries (F(2; 850.9) = 8.95; *p* < 0.001). German psychotherapists had the highest professional experience, differing significantly from CZ (*p* < 0.001) and SK (*p* = 0.007), while CZ and SK did not differ (*p* = 1.000).

### 3.2. RQ 1: Fear of COVID-19 Infection During Psychotherapy in Personal Contact

Fear of COVID-19 infection differed significantly among countries (ME “country”, F(2; 20344.3) = 39.227; *p* < 0.001). Pair-wise post-hoc tests showed highest fear in SK psychotherapists (M = 61.17, SD = 21.48), followed by CZ (M = 51.20, SD = 21.13) and DE (M = 28.89, SD = 25.16) with *p* ≤ 0.005 for all pair-wise post-hoc comparisons. Overall, higher fear of COVID-19 infection during psychotherapy in personal contact was reported by female psychotherapists (M = 47.21, SD = 26.25) than male psychotherapists (M = 39.29, SD = 26.99; F(1; 2793.2) = 5.386; *p* = 0.021); however, no interaction between country and gender was observed (F(2; 36.3) = 0.070; *p* = 0.932). Table 2 summarizes M and SD of the fear of COVID-19 infection for female and male psychotherapists of the studied countries separately.

### 3.3. RQ 2: Changes in Number of Patients Treated

Among all countries, the combined (personal contact + telephone + internet) number of patients treated on average per week during COVID-19 (M = 18.32, SD = 12.86) did not differ from the combined (personal contact + telephone + internet) number of patients treated on average per week in the months before the COVID-19 situation (M = 19.35, SD = 13.73), t(337) = −1.506; *p* = 0.133. Separate analyses for the three countries revealed that while the total number did not change in CZ (t(111) = 1.732; *p* = 0.086), the number increased in DE by on average 12% (t(129) = −2.481; *p* = 0.014) and decreased in SK by on average 25% (t(95) = 3.626; *p* < 0.001).

Table 3 summarizes M and SD of the number of patients treated on average per week in total, and for each treatment format separately for each of the three studied countries and summarizes the results of the *t*-tests.

The number of patients treated on average per week in personal contact decreased from M = 17.84 (SD = 12.02) to M = 9.20 (SD = 10.35) (average decrease 48%, t(337) = 13.224; *p* < 0.001). Separate analyses by country showed an average decrease in CZ by 71% (t(111) = 10.272; *p* < 0.001), in DE by 18% (t(129) = 3.720; *p* < 0.001) and in SK by 76% (t(95) = 11.085; *p* < 0.001).

The number of patients treated on average per week via telephone increased from M = 0.92 (SD = 3.16) to M = 3.28 (SD = 5.22) (average increase 257%, t(337) = −8.717; *p* < 0.001), and the number of patients treated on average per week via internet increased from M = 0.59 (SD = 2.54) to M = 5.83 (SD = 6.82) (average increase 888%, t(337) = −15.346; *p* < 0.001).

In CZ, the number of patients treated on average per week via telephone increased on average by 417%, (t(111) = −6.322; *p* < 0.001), and the number of patients treated on average per week via internet increased on average by 1200%, (t(111) = −10.411; *p* < 0.001). In DE, the number of patients treated on average per week via telephone increased on average by 213%, (t(129) = −3.553; *p* = 0.001), and the number of patients treated on average per week via internet increased on average by 6558%, t(129) = −8.732; *p* < 0.001). In SK, the number of patients treated on average per week via telephone increased on average by 187%, (t(95) = −5.206; *p* < 0.001), and the number of patients treated on average per week via internet increased on average by 343%, (t(95) = −7.625; *p* < 0.001).

For the total sample, the decreases in number of patients treated on average per week by face-to-face psychotherapy in personal contact did not differ from increases in number of patients treated on average per week by remote psychotherapy (telephone + internet): t(337) = −1.506; *p* = 0.133.

The decrease in number of patients treated on average per week in personal contact did not differ from increases in number of patients treated on average per week remotely in CZ (t(111) = 1.732; *p* = 0.086). In DE, the increase in number of patients treated on average per week remotely was higher than the decrease in the number of patients treated in personal contact (t(129) = −2.481; *p* = 0.014), while the opposite was observed in SK (t(95) = 3.626; *p* < 0.001).

### 3.4. Interactions between Changes in the Number of Patients Treated with Treatment Format, Country and Gender

The results of the investigation of interactions between changes in the number of patients treated on average per week (before COVID-19 vs. during COVID-19) with treatment format (personal contact, telephone, internet), country (CZ, DE, SK) and gender are summarized in Table 4.

The total (personal contact + telephone + internet) number of patients treated on average per week during COVID-19 did not differ from the number of patients treated on average per week in the months before the COVID-19 situation (ME “change” F (1; 332) = 1.997; *p* = 0.159), while the change in the total number of patients treated on average per week interacted with country (IE “change × country” F (2; 332) = 5.665; *p* = 0.004), format (IE “change × format” F (1.403; 465.7) = 146.089; *p* < 0.001) and also showed a three-way interaction with country and format (IE “change × format × country” F (2.805; 465.7) = 9.317; *p* < 0.001). A further three-way interaction emerged regarding the changes of patients treated on average per week, the treatment format and gender (IE “change × format × gender” F (1.403; 465.7) = 4.310; *p* = 0.026). The number of patients treated in personal contact, via telephone or internet, differed (ME “format” F (1.374; 456.3) = 302.763; *p* < 0.001). Also, the total number of patients treated differed among countries (ME “country” F (2; 332) = 14.119; *p* < 0.001). Furthermore, the interaction between treatment format and country was significant (IE “format × country” F (2.749; 456.3) = 36.263; *p* < 0.001).

No effect of gender (ME “gender” F (1; 332) = 0.071; *p* = 0.791), as well as no interaction effects between change and gender (IE “change x gender” F (1; 332) = 0.103; *p* = 0.748); format and gender (IE “format × gender” F (1.374; 456.3) = 1.644; *p* = 0.201); country and gender (IE “country × gender” F (2; 332) = 0.897; *p* = 0.409); format, country and gender (IE “format × country × gender” F (2.749; 456.3) = 0.734; *p* = 0.521); change, country and gender (IE “change × country × gender” F (2; 332) = 0.761; *p* = 0.468); and change, format, country, and gender (IE “change × format × country × gender” F (2.805; 465.7) = 0.557; *p* = 0.632) were observed.

For the IE “change × country”, Bonferroni-corrected simple effects tests compared each pair of countries at both time points and revealed that patients treated on average per week before the COVID-19 situation was higher in DE than CZ (*p* = 0.012), but not different between DE and SK as well as between CZ and SK. Total number of patients treated on average per week during COVID-19 was higher in DE than CZ (*p* < 0.001) and SK (*p* < 0.001), but not different between CZ and SK.

For the IE “change × format”, Bonferroni-corrected simple effects tests compared each pair of treatment format at each time point and revealed the following results:Months before COVID-19: The number of patients treated on average per week was higher in personal contact vs. telephone (*p* < 0.001) and vs. internet (*p* < 0.001). Furthermore, the number of patients treated on average per week was comparable for telephone and internet (*p* = 0.573).In the COVID-19 situation: The number of patients treated on average per week was higher in personal contact vs. telephone (*p* < 0.001) and vs. internet (*p* < 0.001). Moreover, the number of patients treated on average per week was higher for internet than for telephone (*p* < 0.001).

The IE “change × format × country” is illustrated in Figure 1. Bonferroni-corrected simple effects tests compared each pair of treatment format at each time point and revealed the following results:
Months before the COVID-19 situation: The number of patients treated in personal contact was higher in DE as compared to CZ (*p* < 0.001) and SK (*p* = 0.021), but did not differ between CZ and SK (*p* = 1.000). The number of patients treated per telephone did not differ among countries (all pair-wise comparisons: *p* = 1.000). The number of patients treated via internet was higher in SK as compared to DE (*p* = 0.025), but did not differ between SK and CZ (*p* = 0.514) as well as between DE and CZ (*p* = 0.422).During the COVID-19 situation: The number of patients treated in personal contact was higher in DE as compared to CZ and SK (*p* < 0.001), but did not differ between CZ and SK (*p* = 1.000). The number of patients treated per telephone or internet did not differ among countries (all pair-wise comparisons: *p* ≥ 0.100).

The IE “change × format × gender” is depicted in Figure 2. Bonferroni-corrected simple effects tests compared female and male psychotherapists for each treatment format at each time point are revealed the following results:
Months before the COVID-19 situation: The number of patients treated on average per week in personal contact (*p* = 0.725), via telephone (*p* = 0.733) and via internet (*p* = 0.918) did not differ in female and male psychotherapists.During the COVID-19 situation: The number of patients treated on average per week in personal contact was higher in male psychotherapists compared to female psychotherapists (*p* = 0.036), while the opposite was observed for psychotherapy via telephone (*p* = 0.015). No difference with respect to gender was observed for the number of patients treated via internet on average per week (*p* = 0.393).

For the ME “format”, Bonferroni-corrected simple effects tests revealed that most patients were treated in personal contact, compared to patients treated via internet (*p* < 0.001) and telephone (*p* < 0.001). Moreover, more patients were treated via internet than via telephone (*p* < 0.001).

The ME “country” revealed a higher number of patients treated on average per week in DE as compared to CZ (*p* < 0.001) and SK (*p* < 0.001), but no difference between CZ and SK.

For the IE “format × country”, Bonferroni-corrected simple effects tests compared each pair of countries and revealed that patients treated in personal contact were higher in DE compared to CZ (*p* < 0.001) and SK (*p* < 0.001), but did not differ between CZ and SK. No differences in patients treated via telephone or internet between countries were observed.

### 3.5. Association between the Fear of COVID-19 Infection and the Number of Patients Treated Per Treatment Format

Pearson correlation analyses revealed a significant negative association between the number of patients treated in personal contact during the COVID-19 situation and the fear of COVID-19 infection (*r* = −0.451, *p* < 0.001). Positive associations were found between the number of patients treated via telephone (*r* = 0.243, *p* < 0.001) and via internet (*r* = 0.197, *p* < 0.001) during the COVID-19 situation and the fear of COVID-19 infection during psychotherapy in personal contact. Also, changes in the provision of psychotherapy in personal contact (*r* = −0.317, *p* < 0.001), via telephone (*r* = 0.225, *p* < 0.001) and via internet (*r* = 0.164, *p* = 0.003) during COVID-19 as compared to the months before COVID-19 correlated with the fear of COVID-19 infection. Thus, a higher fear of COVID-19 infection during psychotherapy in personal contact caused a stronger reduction of psychotherapy provided in personal contact and a stronger increase of psychotherapy provided remotely.

## 4. Discussion

This study revealed that although psychotherapists were confronted with major changes in the way psychotherapy was provided, the number of patients treated could be maintained during the COVID-19 situation when all three countries (CZ, DE, SK) were analyzed together. In total, decreases in psychotherapies in personal contact were compensated by psychotherapies via telephone and internet. Analyses per country revealed that this applied only for CZ, while in DE the increase in the number of patients treated remotely overcompensated the decreased number of patients treated in personal contact, and in SK the opposite was observed. Although psychotherapy in personal contact was reduced and remote psychotherapy increased, psychotherapy in personal contact remained the most frequent psychotherapy format during COVID-19. Changes in the provision of psychotherapy differed among countries and also men and women. In this regard, German psychotherapists showed the lowest decrease in the provision of psychotherapy in personal contact. Female psychotherapists showed a stronger decrease in the number of patients treated in personal contact, while they provided more therapies via telephone compared to their male colleagues. Differences among countries and gender also emerged with respect to the fear of COVID-19 infection during psychotherapy in personal contact, with German and male psychotherapists reporting the lowest fear.

Although the decreases in psychotherapies in personal contact could be compensated by increases in remote psychotherapy averaged among all countries, country-wise comparisons revealed that while in Germany there was even an increase in the number of patients treated, a decrease was observed in SK. One explanation might be differences in legal aspects regarding the use of digital media in psychotherapy. In CZ, remote psychotherapies started to be fully covered by the health insurance companies to psychotherapists that have a contract with some of them during March 2020 until the second half of May 2020. In Germany, limits regarding the amount of remote therapy covered by health insurances was suspended soon after the start of the lockdown up to 1 June 2020, and additional services were reimbursed when conducted remotely. In Slovakia, health insurance companies established remote psychotherapy in only a limited way (as a short email/phone consultation lasting 15 min and costing 4.3 euro, short video consultation lasting 20 min and costing 5.67 euro or a crisis intervention in a single use lasting 45 min and costing 27 euro) and did not establish remote regular psychotherapeutic treatment. Thus, the limited coverage of psychotherapy by health insurance companies might be the reason behind the decrease in the number of patients treated during the COVID-19 situation as compared to the months before in SK. Also, in another study, it was shown that increases in remote psychotherapies did not compensate decreases in psychotherapies in personal contact in Austria [3], where internet-based psychotherapy is rejected by official guidelines. Yet, this discrepancy between the Austrian study [3] and the current study may also be explained by the fact that the Austrian study was conducted in the early weeks of the COVID-19 lockdown whereas the current study on CZ, DE, and SK was conducted when restrictions already were loosened.

Given the general increase in mental health problems during COVID-19 [1,4,5], the provision of mental health care during and after the COVID-19 pandemic is of utmost importance [2]. Therefore, it is important to improve the delivery of remote psychotherapy to enable professional healthcare while reducing the risk of spread of COVID-19 during psychotherapists in personal contact [1,6,7]. Thus, results reveal that the provision of psychotherapy from distance has not lived up to its full potential, as even during the COVID-19 situation, psychotherapy in personal contact remained the most abundant treatment modality.

Overall, it is possible that the responses to this public health emergency might be more than a temporary increase in remote psychotherapy. Since predictions about COVID-19 are largely unclear as of yet, the shift to remote psychotherapy might likely be a longer-term solution of how to continue with mental health care at a safe distance. Some observers already expect that the current COVID-19 crises might lead to a robust shift in the provision of psychotherapy towards digital therapies in the near future [1]. In many countries around the world, including CZ, DE and SK, the outbreak of the COVID-19 pandemic hastened the overcoming of barriers by psychotherapists vis-à-vis remote psychotherapy and seems to be a strong catalyst for the implementation of remote psychotherapy in outpatient psychotherapy. Although the study was conducted in countries that were not severely hit by the COVID-19 pandemic, psychotherapists still adapted towards a remote mental health approach. In view of several advantages of the use of digital technologies for psychotherapy, such as the ability to improve access to mental health care [25], after developing the capabilities of serving patients from distance, psychotherapists might be reluctant to give these up after the COVID-19 situation has ended [1], as the benefits may outweigh its drawbacks, such as technological problems or perceptions of feeling impersonal [9].

An interesting finding of the current study was that German psychotherapists showed the lowest reduction in the number of patients treated in personal contact. In Germany, health insurances loosened regulations with regard to coverage of the costs for psychotherapy via internet [26]. The restriction on the percentage of online psychotherapy per quarter was suspended and additional services were reimbursed, but only when certified providers were used. Thus, while face-to-face psychotherapy was still possible and reimbursed as before, switching to remote psychotherapy necessitated additional efforts on the psychotherapist’s part, such as registering with a certified provider or obtaining informed consent for remote therapy from the patient. In addition, one may hypothesize that the older age of the German sample may have played a role here: possibly older psychotherapists are less experienced with and attracted to digital treatment formats than younger psychotherapists. Taken together, these factors might be responsible for a lower shift from psychotherapy in personal contact to remote psychotherapy as compared to CZ and SK. In contrast, remote psychotherapy was fully covered by the health insurance and also several psychotherapeutic telephone services were provided for free during the outbreak of COVID-19 in the CZ (e.g., Antena, https://psychoterapie.cz/). This could contribute to a higher increase in the number of patients treated remotely in the CZ compared to Germany. Another explanation might be that they experienced less fear of COVID-19 infection during psychotherapy in personal contact, which is supported by the significant associations between the fear of COVID-19 infection and changes in the provision of psychotherapy. This also corroborates the finding that the strongest decrease in psychotherapy provided face-to-face emerged in Slovak psychotherapists, who also showed the highest fear of COVID-19 infection. Differences with respect to fear of COVID-19 infection cannot be explained by higher infection rates in SK and CZ compared to DE. At the start of the online surveys, the cumulative number of confirmed cases was even higher in Germany (2.11 per 100,000 population), compared to CZ (0.74 per 100,000 population) and SK (0.27 per 100,000 population) [27,28]. Similarly, the cumulative number of confirmed deaths related to COVID-19 was highest in Germany (0.096 per 100,000 population), compared to CZ (0.024 per 100,000 population) and SK (0.0048 per 100,000 population) [27,28]. Although speculative, it would be interesting to examine whether differences in media reports about COVID-19 are responsible for differences in fear of infection ratings between countries.

Moreover, the differences in female and male psychotherapists, i.e., the stronger decrease in psychotherapy provided face-to-face in female psychotherapists, corresponds to their higher fear of COVID-19 infection in personal contact. The higher fear of COVID-19 infection in female psychotherapists is in line with previous studies reporting more pronounced emotional distress reactions to stressful events in women [29,30]. However, recent studies in Iranian and Bangladeshi participants observed comparable scores across both genders on a scale specifically developed to assess fear of COVID-19 [31,32], while studies conducted in the Israeli and eastern Europe (Russia and Belarus) population observed a higher rates of fear of COVID-19 in female participants [33,34]. The fear was reported more frequently by women (67%) than men (33%) also when asked to which emotions they felt at the time of the COVID-19 epidemic [21,22]. Thus, differences among gender likely differ among countries.

Overall, female psychotherapists also started to treat more patients via telephone than their male colleagues did, whereas no differences for psychotherapy via internet were found. Previously it has been speculated that attitudes toward specific modes of remote psychotherapy might be moderated by gender [9]; however, further research is required to clearly elucidate differences.

This study has several limitations. One limitation is the cross-sectional design, which might have caused some recall bias regarding the retrospective assessment of the number of patients treated on average per week before the COVID-19 situation. Thus, the psychotherapists’ self-ratings regarding the number of patients treated on average per week likely provides less valid data than analyzing health insurance data would do. Results refer to the first weeks of declining COVID-19 infections and after the participating countries already lifted the COVID-19 lockdowns. Hence, results might differ from the time during the COVID-19 lockdown or some weeks/months later, since fears of COVID-19 and the preferences in the format psychotherapy provided might change dynamically. Further longitudinal studies are required to evaluate the long-term effects of the COVID-19 pandemic on the provision of psychotherapy. A further major limitation is that fear of COVID-19 infection was assessed by a single item measure. Thus, results are not directly comparable to studies using a validated scale to assess the fear of COVID-19 [31]. Another limitation is that psychotherapy via internet comprises quite a broad category, which includes several digital media, such as chat, e-mail and videoconferencing. Furthermore, the online conduct of the survey might have caused a participation of psychotherapists who are in general more used to digital media, and thus might not be representative for all psychotherapists practicing psychotherapy in the respective countries.

## 5. Conclusions

In conclusion, the COVID-19 situation changed the provision of psychotherapy in CZ, DE and SK. While psychotherapy in personal contact was reduced, the provision of psychotherapy via internet increased during the COVID-19 situation in all participating countries. Fear of COVID-19 infection during psychotherapy in personal contact was associated with the decrease in psychotherapy in personal contact and the increase in remote psychotherapy during COVID-19. The reduction of psychotherapy in personal contact was less pronounced in DE, which went along with an increased number of patients treated on average per week during the COVID-19 situation. While the total number of patients treated per week did not differ in CZ, a reduction was observed in SK. Thus, results imply that initiatives in mental health care systems are necessary in SK to cover the need for psychotherapy during and after COVID-19.

## Figures and Tables

**Figure 1 ijerph-17-04811-f001:**
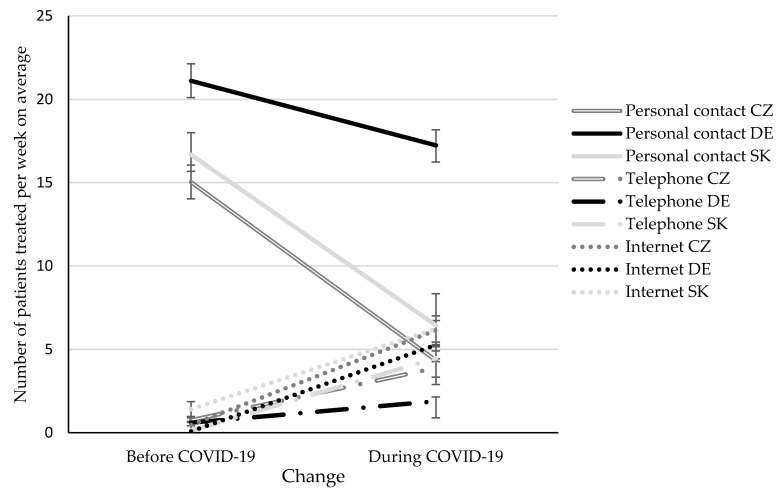
Average number of patients treated per week in personal contact, via telephone or via internet in the months before/during the COVID-19 situation in Czech (CZ), German (DE) and Slovak (SK) psychotherapists. Mean ± standard error.

**Figure 2 ijerph-17-04811-f002:**
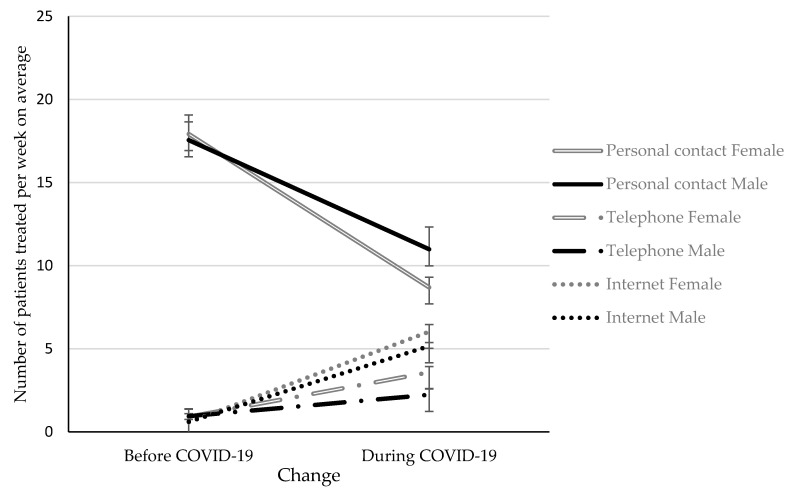
Average number of patients treated per week in personal contact, via telephone or via Internet in the months before/during the COVID-19 situation in female and male psychotherapists averaged among participating countries. Mean ± standard error.

**Table 1 ijerph-17-04811-t001:** Age, gender and years in profession in Czech (CZ), German (DE) and Slovak (SK) psychotherapists.

Variable	Country			Test
	CZ (n = 112)	DE (n = 130)	SK (n = 96)	
Age, M (SD)	44.44 (9.63)	51.45 (10.58)	42.90 (9.61)	ANOVA; *p* < 0.001
Female, %	73.2	77.7	83.3	Chi-square-test; *p* = 0.216
Years in profession, M (SD)	8.55 (8.47)	13.55 (7.37)	9.52 (13.33)	ANOVA; *p* < 0.001

Note: SD = Standard deviation.

**Table 2 ijerph-17-04811-t002:** Fear to become infected with COVID-19 during psychotherapy in personal contact ^1^ with respect to country (Czech Republic (CZ), Germany (DE), Slovakia (SK)) and gender in psychotherapists.

Country		Female			Male	
	N	M	SD	N	M	SD
CZ	82	52.70	19.32	30	47.10	25.35
DE	101	30.61	25.44	29	22.90	23.62
SK	80	62.53	21.51	16	54.38	20.66

^1^ Fear to become infected with COVID-19 during psychotherapy in which psychotherapists are in personal contact with patients was rated on a slider ranging from 0 (“not at all”) to 100 (“extreme”). Note: SD = Standard deviation.

**Table 3 ijerph-17-04811-t003:** Number of patients treated on average per week before and during COVID-19 in the Czech Republic (CZ), Germany (DE) and Slovakia (SK).

Country	Format	Before COVID-19 (M, SD)	During COVID-19 (M, SD)	*t*	*p*
CZ					
(n = 112)	Total	16.26 (12.40)	14.37 (11.44)	1.732	*p* = 0.086
	Personal Contact	15.04 (10.79)	4.33 (6.01)	10.272	*p* < 0.001
	Telephone	0.75 (2.38)	3.88 (5.77)	−6.322	*p* < 0.001
	Internet	0.47 (1.37)	6.15 (6.10)	−10.411	*p* < 0.001
DE					
(n = 130)	Total	21.79 (12.76)	24.38 (12.65)	−2.481	*p* = 0.014
	Personal Contact	21.11 (11.65)	17.24 (10.62)	3.720	*p* < 0.001
	Telephone	0.60 (3.26)	1.88 (2.93)	−3.553	*p* = 0.001
	Internet	0.08 (0.30)	5.26 (6.88)	−8.732	*p* < 0.001
SK					
(n = 96)	Total	19.65 (15.77)	14.71 (11.60)	3.626	*p* < 0.001
	Personal Contact	16.68 (12.92)	4.00 (6.42)	11.085	*p* < 0.001
	Telephone	1.56 (3.71)	4.47 (6.43)	−5.206	*p* < 0.001
	Internet	1.41 (4.42)	6.24 (7.51)	−7.625	*p* < 0.001

Note: SD = Standard deviation.

**Table 4 ijerph-17-04811-t004:** Results of the repeated measures analysis of variance.

Format and Country	Gender	N	Before COVID-19 M (SD)	During COVID-19 M (SD)
Personal contact				
CZ				
	Female	82	15.00 (9.49)	4.11 (5.99)
	Male	30	15.13 (13.94)	4.93 (6.12)
DE				
	Female	101	20.93 (11.59)	16.52 (10.05)
	Male	29	21.72 (12.03)	19.76 (12.27)
SK				
	Female	80	17.11 (13.14)	3.51 (6.10)
	Male	16	14.50 (11.92)	6.44 (7.58)
Telephone				
CZ				
	Female	82	0.74 (2.50)	4.23 (6.34)
	Male	30	0.77 (2.06)	2.93 (3.72)
DE				
	Female	101	0.32 (2.03)	1.82 (2.93)
	Male	29	1.59 (5.74)	2.10 (2.98)
SK				
	Female	80	1.85 (4.00)	5.14 (6.79)
	Male	16	0.13 (0.50)	1.13 (2.19)
Internet				
CZ				
	Female	82	0.29 (0.62)	6.28 (5.86)
	Male	30	0.97 (2.40)	5.80 (6.82)
DE				
	Female	101	0.08 (0.31)	5.60 (7.18)
	Male	29	0.08 (0.26)	4.07 (5.66)
SK				
	Female	80	1.51 (4.77)	6.30 (7.59)
	Male	16	0.94 (1.88)	5.94 (7.31)

Note: SD = Standard deviation; CZ = Czech Republic; DE = Germany; SK = Slovakia; Change = COVID-19 situation vs. months before COVID-19 situation.

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
