# Peer review of "Provision of Psychotherapy during the COVID-19 Pandemic among Czech, German and Slovak Psychotherapists"

_ijerph, 2020, doi:10.3390/ijerph17134811_

Round 1

Reviewer 1 Report

I felt this was a very timely, interesting and well-written article. The context is set out very well, and it is impressive to have conducted a piece of robust research in such a short timeframe. The analysis makes an interesting empirical contribution and findings are discussed appropriately.

I noted only some minor typos/language issues, as follows (line numbers given):

  • 105: the word 'established' doesn't seem to me to be the appropriate word (accepted, funded, reimbursed, supported?)
  • 109: the word 'schools' seems to be missing
  • 123: unnecessary 'to' at start of line
  • 188: unnecessary 'a' preceding paired t-tests
  • 284: 'peer' should be 'per'
  • 314: 'amon' should be 'among'
  • 332: delete duplicate 'in'
  • 469: the word 'fear' is missing
  • 473: use the word 'conduct' rather than 'conduction'

Author Response

Thank you very much for the affirmative review and positive comments! 

We appreciate the thorough review of the manuscript and corrected all typos/ language issues as suggested.

Reviewer 2 Report

This paper discusses the fear of Covid-19 Infection during psychotherapy in personal contact. The title of the paper is confusing and should be better presented.

Inline 119 the authors say "media reports as well as legal restrictions differ among countries, this fear might differ among countries". What leads the authors to this conclusion?

Inline 178. "To evaluate differences in sociodemographic characteristics, univariate ANOVAs and chi-square-tests were conducted." Did the authors consider other methods for data analysis?

The authors must highlight the main challenges in the provision of remote psychotherapy. Also, include suggestions for future works and the study limitations.

Author Response

Authors appreciate the affirmative review and edited the manuscript accordingly.

Regarding the hypothesis that differences in media reports, legal restrictions might affect the fear: this is of course more a speculation than a proven fact. In Slovakia for example, where we observed the highest fear, the prime minister spoke to inhabitants in expressive infantilized manner abetting fear (“we have to eradicate this pest”) and exerted pressure to tighten restrictive measures and urgently and repeatedly warned against breaking the rules. It occasionally had rather absurd consequences, such as a person alone on the street or a lone runner in nature wearing a mask. Besides the recommendations from the Ministry of Health to provide only necessary health care including psychotherapeutic care including psychotherapeutic care.
However, as we cannot directly prove an association between media reports and legal restrictions with the fear of COVID-19 infection, we highlight in the discussion section that this is only speculative and needs to be examined in future studies. 

Regarding statistical analyses: Besides ANOVAs and chi-square-tests additional correlation analyses were added to the revised manuscript.

In the  introduction and discussion, main challenges (i.e., legal limitations, technological problems, perceptions of feeling inpersonal, difficulties in developing a therapeutic alliance or reach an accurate diagnosis) of remote psychotherapy are highlighted. Also, suggestions for future works (associations between media reports and fear of COVID-19 infections, further longitudinal studies to investigate long-term effects of COVID-19 on provision of psychotherapy) and limitations (cross-sectional design, only self-reports, no validated instrument to measure fear of COVID-19 infection, no refinement of the category of psychotherapy via internet, online conduction might cause selection bias) are included.

Reviewer 3 Report

The ms. is a contribution regarding the fear of psychotherapists to become infected with COVID-19 during psychotherapy in personal contact and assessed how the provision of psychotherapy changed due to the COVID-19 pandemic.

The manuscript is well organized, anyway I suggest these changes:

  • Review the Results section

The layout of tables is not very readable.

  1. Table 1: replace “statistics” with “test” reporting in column only p values. Please report in the caption which kind of statistics was used (anova, chi squared).
  2. Table 2: please remove statistics’ columns. Report statistics only in the text.
  3. Table 3: please remove the term statistics, and report in the table headings t and p with the corresponding values in column.
  4. Table 4 is not readable. I suggest reorganizing the paragraph 3.4 in order to include in the text the effects reported in the table.

Please check some typos in the text.

Author Response

Thank you for the positive review.

The results section was changed according to the useful comments.

We apologize for the typos and edited the manuscript.

Reviewer 4 Report

Thanks for your submission. This is an interesting study looking at the fear of COVID-19 infection during psychotherapy in personal contact and changes in provision of psychotherapy among Czech, German and Slovak psychotherapists. I found the submission useful.

However, the title can of the paper can be simpler rather than a four liner. It will be better received by the readership if the title of the paper can be simplified. 

You can improve the introduction to include some other global citations on the subject and add few more sentences to the conclusion. 

Author Response

Authors appreciate the affirmative review and edited the manuscript accordingly.

The title was shortened considerably and the conclusion was extended.

Reviewer 5 Report

This is an important and timely study. The results of this study provide new knowledge to the field of psychotherapy in pandemic.

My first suggestion for the authors is to shorten the topic into “Provision of Psychotherapy during COVID-19 pandemic among psychotherapists in Czech, German and Slovak.”

Second, the correlation between fear of COVID-19 infection and changes in provision of psychotherapy should be examined.

Author Response

Authors appreciate the positive review and edited the manuscript according to the comments.

The title of the manuscript was shortened and modified according to the reviewers`suggestion.

Thank you for the suggestion to run correlation analyses between fear of COVID-19 infection and changes in provision of psychotherapy. We added this aspect in the revised manuscript.